# Atypical Hemolytic Uremic Syndrome after SARS-CoV-2 Infection: Report of Two Cases

**DOI:** 10.3390/ijerph191811437

**Published:** 2022-09-11

**Authors:** Iwona Smarz-Widelska, Małgorzata Syroka-Główka, Joanna Janowska-Jaremek, Małgorzata M. Kozioł, Wojciech Załuska

**Affiliations:** 1Department of Nephrology, Cardinal Stefan Wyszynski Provincial Hospital, 20-718 Lublin, Poland; 2Chair and Department of Medical Microbiology, Medical University of Lublin, 20-093 Lublin, Poland; 3Department of Nephrology, Medical University of Lublin, 20-954 Lublin, Poland

**Keywords:** aHUS, TMA, eculizumab, COVID-19

## Abstract

Atypical hemolytic uremic syndrome (aHUS) is a life-threatening disease causing systemic thrombotic microangiopathy (TMA) due to the fact of complement dysregulation. Immune activation by viruses, including SARS-CoV-2, can lead to the development of an episode of aHUS against a background of genetic dysregulation in the complement pathway. This paper presents an analysis of two cases of aHUS—siblings diagnosed with familial disease, with a genetic predisposition to aHUS, in whom infection with SARS-CoV-2 was a strong trigger of disease recurrence. The quick recognition and treatment with eculizumab in the early stage of the disease resulted in a rapid improvement in clinical conditions and laboratory parameters.

## 1. Introduction

Atypical hemolytic uremic syndrome (aHUS) is a very rare disorder in which life-threatening thrombotic microangiopathy (TMA) shows up as a dysregulation of the alternative pathway (AP) of complement activation. It is characterized by endothelial dysfunction, microvascular thrombosis, tissue ischemia, and organ damage. A medical triad includes microangiopathic hemolytic anemia (MAHA), thrombocytopenia, and acute kidney injury (AKI). Primary aHUS is associated with gene mutation of complement-regulatory proteins, factor H (FH), MCP/CD-48, and factor I (FI), or with changes in complement-activating components, C3 and factor B (FB). The secondary form is related to clinical conditions such as cancer, systemic diseases, malignant hypertension, pregnancy, infections, or medications [1,2,3]. The constantly evolving COVID-19 pandemic has led to many acute and chronic clinical complications of which the pathophysiology is not fully understood [4]. A review of the literature shows that the damaging effect of the SARS-CoV-2 virus may be due to direct action by the pathogen on endothelial cells, as was demonstrated in a postmortem study [5] and/or the result of inflammatory response. The last one promotes hypercytokinemia, increased inflammation leading to endothelial dysfunction, and thrombus formation [6,7]. There are compelling arguments that support the concept that overactivation of the complement pathway triggers hyperinflammation and thrombosis [8]. From postmortem research evidence, it is known that there exist complement fragments in alveolar endothelium and kidney tissue [9]. Although the pathogenesis of multiorgan damage in SARS-CoV-2 differs from aHUS, uncontrolled activation of the complement pathway plays a key role in both diseases. It is associated with the formation of the final products of the complement cascade, responsible for the pro-inflammatory, prothrombotic effects, and cell apoptosis [10,11]. In the past, the standard of care for aHUS patients included plasma infusion (PI) or plasma exchange (PE). This treatment was unsuccessful in most cases, resulting in end-stage renal disease (ESRD) development after the first episode of aHUS or within a few years [12]. A breakthrough therapy drug was eculizumab (Ecu), a humanized monoclonal antibody that blocks the cleavage of the final complement protein, C5, into proinflammatory C5a and C5b as well as the formation of C5b-9 responsible for cell lysis, inhibiting complement-mediated thrombotic microangiopathy (TMA) [13,14,15,16].

The presented work aimed to describe, for the first time, the relapse of familial aHUS in two related patients with a heterozygous mutation of MCP/CD46 in the complement system in whom the last relapse was triggered by SARS-CoV-2 infection.

## 2. Presentation of Cases

We report the clinical findings in two patients: brothers (19 and 23 years old) with atypical HUS relapse during SARS-CoV-2 infection.

Important background: Patients had a heterozygous mutation of MCP/CD46 in the complement system. Genetic analysis, performed at the Institute of Human Genetics University Hospital of Cologne, showed in exon 2 of the CD46 gene in two males (brothers) and the mother a heterozygous c.202_203dupTA frameshift mutation. This mutation results in a frameshift and premature stop codon at position 107 of the amino acid (p.Phe69Thrfs*39). Sequence analysis of exon 2 of the CD46 gene on the patient’s father’s DNA revealed no pathology.

### 2.1. Patient No. 1

A 19 year old male was diagnosed with aHUS at the age of 14. Thus far, the patient has had two relapses of mild disease manifestation at 14 and 16 years of age. The patient was then treated with a concentration of platelets, red cells mass (RCM), and fresh frozen plasma (FFP). Both outbreaks were associated with an infection of the upper respiratory tract. In October 2021 (at the age of 18), the patient came to the district hospital in their place of residence. He complained of fever, cough, headaches, and pain in the lumbar region with accompanying hematuria and oliguria. An RT-PCR test for SARS-CoV-2 was performed and gave a positive result. According to the analysis, the patient was admitted to the Infectious Diseases Department. On admission, he was generally in a mild health condition, with circulatory efficiency (RR: 125/95 mmHg; HR: 70/min) and respiratory efficiency (99% saturation without oxygen supplementation), yellowing skin layers, and without signs of hemorrhagic diathesis. Laboratory tests revealed thrombocytopenia (PLT: 27,000/µL); increased level of d-dimers > 18,000 ng/mL; markers of hemolytic anemia: (bilirubin level: 6.1 mg/dL; LDH: 1136 U/L); decreased hemoglobin level (Hgb: 12.1 g/dL); elevated creatinine level: 2.2 mg/dL. Additionally, markers of inflammation were found to be elevated (CRP: 38.4 mg/L; PCT: 0.52 ng/mL; IL-6: 21.5 pg/mL), and hematuria and proteinuria were marked. In the assessment of a chest CT (computed tomography) scan, he had no lung involvement, with a normal chest. The treatment process included empiric antibiotic therapy (ceftriaxone, i.v.), antiviral therapy (remdesivir, i.v.), and FFP. During the hospitalization period, worsening anemia (Hgb: 9.1 g/dL) and thrombocytopenia (PLT: 13,000/µL) were observed. The patient was diagnosed with another relapse of aHUS and was transferred to a specialist center on the fifth day of hospitalization.

The patient was admitted to the Nephrology Department of the Provincial Hospital in a generally good and stable condition. Tests revealed hemolytic anemia (Hgb: 8.3 g/dL; a low haptoglobin level; LDH exceeding 10 times the normal limit); severe thrombocytopenia (PLT: 16,000/µL); elevated creatinine (1.83 mg/dL; eGFR 52.7 mL/min/1.73 m^2^); proteinuria: 300 mg/dL; elevated level of transaminases (AST: 146 U/L; ALT: 42 U/L). Studies showed a decreased level of the complement C3 component (0.8 g/L), a normal level of C4, and an increased level of total complement activity CH 50 (346.3 EqU/mL). Immediately, treatment was started using a C5 blocker (eculizumab; dose: 900 mg, i.v.).

On the next day of hospitalization, after the first dose of eculizumab, the patient experienced epigastric pain, nausea, and vomiting, which increased after a meal. The repeated test revealed an increased level of transaminases (AST: 522 U/L; ALT: 461 U/L); bilirubin (5.62 mg/dL); GGTP 756 U/L); alkaline phosphatase (ALP: 214 U/L). An abdominal CT scan provided information regarding moderate liver and spleen enlargement and moderate dilatation of the intrahepatic bile ducts. To extend the investigation, MRI (magnetic resonance imaging) was performed, which showed moderately dilated intrahepatic and extrahepatic ducts. In the distal part of the common bile duct and the gallbladder, sludge was observed. The patient received conservative treatment (i.e., antispasmodics, proton pump inhibitor (PPI), ursodeoxycholic acid (UDCA)).

A gradual resolution of the symptoms and a downward trend in the cholestasis indicators were observed. Seven days after the first dose of eculizumab, the platelet count and creatinine levels improved (PLT: 212,000/µL; creatinine: 1.28 mg/dL; eGFR: 81.2 mL/min/1.73 m^2^), while there was still anemia (Hgb: 7.5–7.9 g/dL). Two weeks after the first dose, normalization of renal function markers was observed (creatinine: 0.86 mg/dL; eGFR: 126 mL/min/1.73 m^2^) among others: LDH (342 U/L) and haptoglobin (0.88 g/L). After the second dose, the patient was discharged home.

After one month of treatment with eculizumab, the parameters of liver function returned to normal. The total complement activity of CH 50 significantly decreased (3.2 EqU/mL). After 8 weeks of treatment, the morphology returned to normal.

In the above case, a 19 year old patient was admitted in October 2021 (*Patient no. 1*). Due to the course of the disease, his older brother (*Patient no. 2*) came to a specialist center in January 2022 with clinical symptoms suspecting of being an aHUS attack (preliminary diagnosis during childhood).

### 2.2. Patient No. 2

A 23 year old male was diagnosed with aHUS at the age of seven. Thus far, the patient had three relapses of mild disease manifestation at 7, 8, and 21 years of age. The patient was then treated with a concentration of platelets, RCM, and FFP. All relapses were related to an upper respiratory tract infection. Currently, the patient was admitted to the Emergency Department due to the presence of fever, weakness, and hematuria. An RT-PCR test for SARS-CoV-2 infection was positive. A performed chest X-ray revealed the possibility of fine speckled peribronchial densities around the recesses. The patient was admitted to the Infectious Diseases Department. On admission, he was feeling well and was circulatory and respiratory efficient. Laboratory tests revealed thrombocytopenia (PLT: 12,0000/µL); an increased level of d-dimers: 3816 ng/mL; markers of hemolytic anemia (bilirubin level: 6.44 mg/dL; LDH: 1383 U/L; decreased haptoglobin level: 0.01 g/L); decreased level of hemoglobin (Hgb: 11.5 g/dL); increased creatinine (1.61 mg/dL; eGFR: 59.4 mL/min/1.73 m^2^); proteinuria: 300 mg/dL; increased transaminases (AST: 101 U/L; ALT: 56 U/L). Complement fractions of C3 and C4 were normal, and we noted moderately elevated inflammatory markers (CRP: 37.2 mg/L). The patient was diagnosed with another aHUS relapse and qualified for treatment with eculizumab.

One week after the first dose of eculizumab, significant changes in the lab results were observed as were normalized parameters: platelets (PLT: 391,000/µL); haptoglobin (0.84 g/L); LDH (287 U/L); transaminase levels and kidney function (creatinine: 1.1 mg/dL; eGFR: 94.1 mL/min/1.73 m^2^); increased hemoglobin (Hgb: 12.5 g/dL). According to the good response to treatment, the patient was discharged home.

Two weeks after the first dose, normalization of hemoglobin levels (Hgb: 14.0 g/dL) and resolution of proteinuria were observed.

It is worth noting that both of the patients were under control and continued treatment with eculizumab, and their parameters were measured during the treatment.

In Table 1 and Table 2, we present how the laboratory findings of the patients changed over the weeks of eculizumab administration.

## 3. Discussion

There is some evidence that overactivation of the complement system in COVID-19 results in several clinical manifestations such as thrombotic microangiopathy, thrombocytopenia, kidney damage, and thrombophilia. Complement is a key player in the immune response to infectious agents, including viruses, but its abnormal activation plays a critical role in promoting inflammation. Complement can be activated by the classical (CP), lectin (LP), and alternative (AP) pathways. SARS-CoV-2 directly activates the lectin/alternative complement pathway in the lungs, microcirculation, and kidneys, which leads to the generation of potent anaphylatoxins (ATs): factor C3a and C5a [17,18]. C5a is formed as a result of the decomposition of the C5 molecule. This mediator is responsible for exacerbation of the inflammatory process and the prothrombotic effect. C5b is a second cleavage product of C5, which is involved in the formation of lytic membrane attack complexes (MACs, C5b-9), causing cell membrane rupture and lysis [7].

In a recently published research paper, *Aiello* et al. showed that in COVID-19, as in aHUS, the C5a/C5aR axis is a prothrombotic effector that promotes platelet aggregation and thrombus formation in the microcirculation [19]. Previous literature has shown that aHUS recurrence can be induced by infection agents including viral pathogens such as influenza A (H1N1 virus) and influenza B [20,21]. The link between SARS-CoV-2/COVID-19 and aHUS has been documented as well but in very few cases. *Ville* et al. described the recurrence of aHUS in a 28 year old female patient in whom genetic analysis revealed the presence of a pathological heterozygous variant in the MCP (membrane cofactor protein) [22]. In addition, *Kaufelld* et al. discussed two cases of patients with a pathogenic factor H mutation in whom COVID-19 triggered the first aHUS attack [23]. The cases that we reported in our paper confirm that SARS-CoV-2/COVID-19 has joined the list of potential aHUS triggers. The powerful activation of the complement provoked by SARS-CoV-2, in the context of an inborn complement control deficiency, which was associated with the MCP mutation, led to overapplication of the alternative complement pathway, resulting in thrombotic microangiopathy. Our findings indicate that despite an even mild course of COVID-19, without pulmonary manifestation and with a moderate increase in inflammatory parameters, it can develop into aHUS relapse. In both analyzed cases, we observed a very high degree of thrombocytopenia, high levels of d-dimers, markers of increased hemolysis—high LDH, low haptoglobin, a decrease in hemoglobin, and a moderate increase in creatinine levels. We suggest that the severe thrombocytopenia, following the platelets used for microclotting, was a consequence of SARS-CoV-2 enhancement of the prothrombogenic effect dependent on the C5a/C5aR axis. In our observations, the first brother had a much more severe relapse. The clinical manifestations, performed diagnostics, and laboratory tests showed signs of extrahepatic cholestasis. At first, as a reason for the patient’s condition, the side effects of remdesivir and/or eculizumab administration were considered. Due to the gradual cholestasis regression in the continuation of eculizumab treatment, an iatrogenic complication was excluded. In our opinion, the cause of cholestasis was increased hemolysis, which led to microangiopathic hemolytic anemia. On the other hand, the second described patient, who came directly to the hospital and received the first dose of eculizumab, did not develop such severe hemolysis, which shows the need to implement treatment as soon as possible.

Eculizumab is a humanized monoclonal antibody that has revolutionized the treatment of patients with aHUS. The last decade is based on numerous prospective clinical trials, which indicate the efficacy and safety of Ecu treatment [24,25,26]. Previous reports have shown that early initiation of Ecu treatment is the best possible chance for renal function recovery and prevents disease progression and life-threatening TMA complications [13,14,15]. In the presented cases, thanks to the use of eculizumab at an early stage of disease relapse, renal dysfunction was completely restored, and the hematological parameters improved. The duration of treatment with eculizumab in our patients will be considered individually, taking into account the potential risks and benefits of treatment.

## 4. Conclusions

To sum up, patients with constitutional defects of aHUS should be aware of the potential factors that can cause recurrence. The key is to seek medical attention and start treatment immediately in order to stop the course of the disease.

## Figures and Tables

**Table 1 ijerph-19-11437-t001:** Laboratory parameters of patient no. 1 over time after drug administration.

Parameter	Week							
	0	I	II	III	IV	VI	VIII	X
Creatinine (0.7–1.2 mg/dL)	2.0	1.28	0.86	0.87	0.78	0.75	0.71	0.73
eGFR (mL/min/1.73 m^2^)	46	81	126	126	131	134	137	135
RBC (4.5–5.9·10^6^/µL)	3.19	2.7	3.1	3.4	3.8	4.4	4.8	4.9
HGB (14–18 g/dL)	9.1	7.7	9.3	10.1	11.2	12.9	13.9	14.4
PLT (150–400·10^3^/µL)	13	212	153	169	269	292	248	219
LDH (<480 U/L)	1131	570	342	251	158	142	144	223
Total bilirubin (0.2–1 mg/dL)	4.0	4.9	2.0	1.42	1.0	0.71	-	-
Haptoglobin (0.3–2 g/L)	0.01	0.03	0.88	1.37	2.57	2.41	1.39	1.09
CH50 (70–187 EqU/mL)	346	<1.0	3.2	5.7	25.5	-	5.3	-

**Table 2 ijerph-19-11437-t002:** Laboratory parameters of patient no. 2 over time after drug administration.

Parameter	Week							
	0	I	II	III	IV	VI	VIII	X
Creatinine (0.7–1.2 mg/dL)	1.6	1.1	-	1.03	1.0	0.99	0.94	1.03
eGFR (mL/min/1.73 m^2^)	59	94	-	102	97	107	114	102
RBC (4.5–5.9·10^6^/µL)	4.0	4.5	4.7	5.1	5.2	5.5	5.6	5.8
HGB (14–18 g/dL)	11.5	13.0	14,0	14.9	15.5	16.2	16.8	17.3
PLT (150–400·10^3^/µL)	12	391	277	196	195	253	247	257
LDH (<480 U/L)	1383	287	190	168	171	148	167	142
Total bilirubin (0.2–1 mg/dL)	6.4	1.3	-	1.46	-	1.68	1.12	1.72
Haptoglobin (0.3–2 g/L)	0.01	0.84	1.05	1.19	1.12	1.08	0.98	1.07
CH50 (70–187 EqU/mL)	81.5	-	7.8	2.3	2.3			

## Data Availability

The datasets used and/or analyzed in the current study are available from the corresponding author on reasonable request.

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
