# Peer review of "Atypical Hemolytic Uremic Syndrome after SARS-CoV-2 Infection: Report of Two Cases"

_ijerph, 2022, doi:10.3390/ijerph191811437_

Round 1

Reviewer 1 Report

The cases of this paper are quite outstanding as they contribute to have knowledge about an effective treatment for atypical HUS.

The cases are well presented. However, as the article is only about the two cases I have doubt that it could be considered as an article. It would be more suitable to consider it as a case report.

Maybe the explanation about the mechanism of action of eculizumab should be explained in the introduction instead of in the discussion

There is a mistake in Table 2 description, as it refers it to be data of patient 1 instead of patient 2

Author Response

The authors sincerely thank the Reviewer for all comments and critical assessments of our paper.

We have done our best to improve it. Below are our responses to each concern.

Reviewer 2 Report

Dear authors,

I have read with interest your report of two aHUS cases. The manuscript offers an interesting insight on these two peculiar clinical presentations.

Please, find here my comments to improve the paper.

To reach more readers, the title would better read “Atypical Hemolytic Uremic Syndrome Relapse after SARS-CoV-2 Infection: Report of two cases”.

Please, also change the study type from Article to Case Report. Also, being a report of cases, the text should not follow the structure of a original paper (with methods, results, etc.). “materials and Methods” can be moved combined with introduction

Line 14 “infection” could be deleted.

Line 17 change “coronavirus” to “SARS-CoV-2” or, if the infection was symptomatic, to “COVID-19”

Line 39 “Previous studies suggest that some clinicopathological characteristics of COVID-19 are common to aHUS.” I suggest to rephrase this sentence and cite appropriate references. We still know a little on COVID-19 pathophysiology, and it would be undue to compare with aHUS.

Line 49 delete “coronavirus”. “CoV” in “SARS-CoV-2” means ‘coronavirus’

Line 144: what is the specific “convincing” evidence?

Line 185 “This fact confirms the need to implement treatment as soon as possible.” You cannot conclude this on the basis of a single experience. If strong evidence exists, please cite. If not, rephrase considering this as a hypothesis. Although a valid one.

Author Response

(The authors gave the same response as above.)

Round 2

Reviewer 2 Report

Authors have fully addressed all my concerns